# Telerehabilitation by Videoconferencing for Balance and Gait in People with Parkinson’s Disease: A Scoping Review

**DOI:** 10.3390/geriatrics9030066

**Published:** 2024-05-23

**Authors:** Carla Silva-Batista, Filipe Oliveira de Almeida, Jennifer L. Wilhelm, Fay B. Horak, Martina Mancini, Laurie A. King

**Affiliations:** 1Department of Neurology, Oregon Health and Science University, Portland, OR 97239, USA; batistac@ohsu.edu (C.S.-B.); wilhelmj@ohsu.edu (J.L.W.); horakf@ohsu.edu (F.B.H.); mancinim@ohsu.edu (M.M.); 2Exercise Neuroscience Research Group, University of São Paulo, São Paulo 05508-070, Brazil; filipe.emanuel@hotmail.com

**Keywords:** Parkinson’s disease, telerehabilitation, videoconferencing, remote, mobility

## Abstract

Although supervised and real-time telerehabilitation by videoconferencing is now becoming common for people with Parkinson’s disease (PD), its efficacy for balance and gait is still unclear. This paper uses a scoping approach to review the current evidence on the effects of telerehabilitation by videoconferencing on balance and gait for patients with PD. We also explored whether studies have used wearable technology during telerehabilitation to assess and treat balance and gait via videoconferencing. Literature searches were conducted using PubMed, ISI’s Web of Knowledge, Cochrane’s Library, and Embase. The data were extracted for study design, treatment, and outcomes. Fourteen studies were included in this review. Of these, seven studies investigated the effects of telerehabilitation (e.g., tele-yoga and adapted physiotherapy exercises) on balance and gait measures (e.g., self-reported balance, balance scale, walking speed, mobility, and motor symptoms) using videoconferencing in both assessment and treatment. The telerehabilitation programs by videoconferencing were feasible and safe for people with PD; however, the efficacy still needs to be determined, as only four studies had a parallel group. In addition, no study used wearable technology. Robust evidence of the effects of telerehabilitation by videoconferencing on balance and gait for patients with PD was not found, suggesting that future powered, prospective, and robust clinical trials are needed.

## 1. Introduction

Parkinson’s disease (PD) is one of the most common movement disorders and represents the second most common degenerative disease of the central nervous system. Balance and gait difficulties are common motor symptoms in PD and are major risk factors for falls and institutionalization [1,2,3,4]. Meta-analyses have demonstrated that physical exercise is an important form of treatment to alleviate balance and gait disturbances in PD [5,6]. However, during the coronavirus disease 2019 (COVID-19) pandemic, millions of people were forced to self-isolate and halt regular exercise. People with PD experienced worsening motor symptoms, including balance and gait issues, which contributed to increased falls [7]. One method to combat this decline in mobility during periods in which in-person rehabilitation is not available could be the use of telerehabilitation, which is now becoming more common for people with PD [8,9,10,11,12].

Telerehabilitation is defined as a healthcare team’s use of communications at a distance to remotely provide rehabilitation services [13,14]. Telerehabilitation provides an opportunity for timely and accessible services in the homes of people with PD [15]. There are different telerehabilitation technology forms but the most common are videoconferencing, messaging, email, and web-based platforms [15]. There are also more advanced technologies including virtual reality, whereby a physical and/or cognitive experience with multiple sensory inputs (e.g., visual, auditory, and somatosensory) is used to create environments that challenge a person’s balance. Although virtual reality is the telerehabilitation intervention most commonly used in the literature for people with PD [6,16,17], it is primarily used only in research [18], and with the exception of two studies [16,19], it is not typically used with videoconferencing. Telerehabilitation via phone or videoconferencing without virtual reality is considered a basic technology [15] and accessible to most people [20,21]. Two systematic reviews have investigated the costs associated with telerehabilitation [22,23]. Kairy et al. [22] demonstrated that lower costs for healthcare arose from telerehabilitation. Del Pino et al. [23] found that telerehabilitation can be less costly and burdensome than in-person rehabilitation performed in the clinic for people with neurological and cardiovascular diseases. Although telerehabilitation delivered remotely shows lower costs compared with face-to-face rehabilitation at home [24], it is still unknown if telerehabilitation is as effective and safe as facility-based rehabilitation, especially for balance and gait impairments. Overall, home-based telerehabilitation by videoconferencing is a promising solution that may provide an opportunity for timely, possibly effective, and accessible services to a large base of patients, especially in low-resource, rural areas.

In the current review, we will investigate the effects of supervised, home-based, real-time videoconferencing telerehabilitation on balance and gait outcomes. Successful telerehabilitation programs by videoconferencing may require the ability to also assess balance and gait outcomes remotely, although we have found no evidence in the literature that the same balance and gait outcomes, assessed remotely and in person, provide the same information. Balance and gait are difficult to assess by videoconferencing because it is not clear we can evoke imbalance remotely (e.g., postural perturbations, unstable surfaces) and capture multiple domains of balance and gait objectively while keeping patients safe like in-person, supervised assessments. 

A few studies have used supervised, home-based, real-time videoconferencing for both the assessment of balance and the gait of people with PD [8,10,19,25,26,27,28]. These studies have investigated the effects of supervised, home-based, real-time videoconferencing telerehabilitation on self-reported balance (Activities-Specific Balance Confidence scale—ABC), balance scales (Berg Balance Scale—BBS), motor symptoms (Movement Disorder Society-Sponsored Revision of the Unified Parkinson’s Disease Rating Scale part III (MDS-UPRDS-III)), walking speed (6 m walk, 10 m walk, 2 min walk), timed-up-and-go tests (TUG), and five times sit-to-stand assessed by videoconferencing [8,10,19,25,26,27,28]. However, there has been no review paper summarizing the efficacy of telerehabilitation for balance and gait by videoconferencing, or providing overall guidance on how we can implement a fully remote model (i.e., both assessment and treatment) by videoconferencing. Furthermore, it is unclear whether full telerehabilitation by videoconferencing is appropriate for all patients with PD (e.g., across disease severity, cognitive impairments, age, fall risk, etc.). Finally, while wearable sensors are emerging as a means of quantifying objective balance and gait impairments in PD [29,30,31,32,33,34] and can provide real-time feedback to therapists based on objective balance and gait metrics during in-person rehabilitation programs [35,36,37,38], the use of this approach is unexplored for telerehabilitation by videoconferencing.

Therefore, our study aimed to review the current evidence on the effects of telerehabilitation by videoconferencing on the balance and gait of patients with PD. We also explored whether studies have used wearable technology in telerehabilitation programs in both assessment and treatment by videoconferencing.

## 2. Materials and Methods

The literature searches were conducted in the following four computerized databases from the earliest record up to March 2023: PubMed, ISI’s Web of Knowledge, Cochrane’s Library, and Embase. The inclusion criteria were (a) a population with a diagnosis of idiopathic PD; (b) any study design (e.g., quasi-experimental, experimental, case study) published in a peer-reviewed journal and available in full text; and (c) balance and gait outcomes assessed before and after supervised, home-based, real-time videoconferencing telerehabilitation interventions such as physical exercise and virtual reality. The exclusion criteria were telerehabilitation with the use of robotic devices, telerehabilitation without videoconferencing (e.g., physical exercise and/or virtual reality without supervision in real-time), and telerehabilitation studies that did not include gait and/or balance outcomes.

The search was limited to the English language. All the identified and retrieved electronic search titles, selected abstracts, and full-text articles were independently evaluated by two of the authors (FOA and CSB) to assess their eligibility. In the case of disagreements, a consensus was adopted or, if necessary, a third reviewer evaluated the article (JW). The search strategy used a combination of Medical Subject Heading terms and keywords such as Parkinson’s disease, Telerehabilitation, Remote rehabilitation, Virtual rehabilitation, Virtual assessment, and Remote assessment. The PICOS (Population, Intervention, Comparison, Outcome, and Study) strategy was used to investigate our objective, as follows:

P = Parkinson’s disease, Parkinson’s;

I = Telehealth, telerehabilitation, telerehabilitation, remote rehabilitation, virtual rehabilitation, videoconferencing;

C = Not using telerehabilitation, face-to-face intervention, home-based rehabilitation;

O = Balance, gait, walking, posture, motor symptoms, mobility;

S = Review, systematic review, meta-analysis, case–control, feasibility, randomized controlled trial, pilot study, quasi-experimental.

## 3. Results

### 3.1. Study Selection

To investigate the current evidence on the effects of telerehabilitation on balance and gait delivered by videoconferencing for PD, the search identified 2601 articles retrieved from computerized databases up to September 2023 (PubMed, ISI’s Web of Knowledge, Cochrane’s Library, and Embase). After removing duplicate items (779), 1822 remained, in which titles and abstracts were read and 104 were selected. Of these, 72 were excluded and 32 were fully assessed for eligibility; however, only 14 studies met the study criteria, which were included in the final analysis. The search process is depicted in Figure 1.

### 3.2. Study Characteristics

Fifteen studies did not use telerehabilitation by videoconferencing [39,40,41,42,43,44,45,46,47,48,49,50,51,52,53] and three studies did not investigate the effects of telerehabilitation by videoconferencing on gait and balance outcomes [54,55,56]. The search identified 14 studies that used telerehabilitation by videoconferencing aiming to alleviate balance, gait, and motor symptoms (MDS-UPDRS-III) disturbances (Table 1) [8,10,16,19,25,26,27,28,57,58,59,60,61,62]. Of these studies, only four were randomized controlled trials (RCTs) [16,57,58,62], eight were non-RCTs [8,10,25,26,27,28,60,61], and two were a case–control [19,59]. 

A variety of interventions were used, such as adapted physiotherapy exercises in standing and sitting positions [8,27], yoga [25,26], adapted and safe Argentine tango classes [58], dance (e.g., jazz, tap, samba, forró, salsa, and tango) [28], balance-virtual reality training [16], treadmill-virtual reality training [19], mobility and postural transition exercises [10], exercises of low-intensity static and dynamic postural control integrated with breathing patterns [59], motor rehabilitation (e.g., flexibility, balance, and gait training) [57,60,61], and Lee Silverman Voice Treatment^®^ BIG (LSVT^®^ BIG) [62]. The number of telerehabilitation sessions ranged from 8 [25] to 80 [27] and sample sizes ranged from 2 [19,59] to 86 people with PD [27].

Although 14 studies [8,10,16,19,25,26,27,28,57,58,59,60,61,62] used telerehabilitation by videoconferencing for PD, of these, only 7 studies [8,10,19,25,26,27,28] used videoconferencing to both assess and treat balance and gait, as demonstrated in Table 1. Although no study has investigated the implementation of wearable technology into telerehabilitation programs to assess and treat balance and gait remotely, we also created a search term to evaluate this potentially effective approach. Finally, measures related to balance and gait, such as freezing of gait (FOG), motor symptoms (MDS-UPDRS-III), five times sit-to-stand, and mobility (TUG test) are also included in this scoping review paper.

### 3.3. Effects of Telerehabilitation by Videoconferencing on Balance and Gait Assessed by Videoconferencing

Of the 14 studies included in this scoping review (Table 1), only 7 studies both assessed and treated people with PD by videoconferencing the telerehabilitation programs [8,10,19,25,26,27,28].

The effects of tele-yoga [25], treadmill-virtual reality training [19], and dance [28] on ABC scores were investigated. These studies showed the feasibility and safety (no adverse events) of 8 sessions (90 min each session), 12 sessions (15–60 min each session), and 16 sessions of tele-yoga (*n* = 8), treadmill-virtual reality training (*n* = 2), and dance (*n* = 12) in people with PD, respectively. Only treadmill-virtual reality training [19] and dance (e.g., jazz, tap, samba, forró, salsa, and tango) [28] showed self-reported balance improvement in people with mild-to-moderate PD; however, since both studies used a quasi-experimental design, the positive effects on ABC scores are still unclear. 

Kwok et al. [25] investigated the effects of eight sessions (90 min each session) of tele-yoga on the Freezing of Gait Questionnaire (FOGQ) scores but no significant effect was observed in eight people with moderate PD. Tardelli et al. [27] compared the effects of 80 sessions (60 min each session) of telerehabilitation with adapted physiotherapy exercises in standing and sitting positions (*n* = 57) with the control intervention with exercise (*n* = 29) on the New Freezing of Gait Questionnaire (NFOGQ) scores in people with mild-to-severe PD. Both groups reported worse self-reported FOG after the interventions. These results suggest that tele-yoga and telerehabilitation with adapted physiotherapy exercises in standing and sitting positions are not effective in decreasing self-reported FOG. A systematic review has shown that generic exercises, such as yoga and physiotherapy not aimed at FOG, are not as effective in decreasing FOG severity compared to interventions aimed directly at alleviating FOG [63]. Thus, future studies should include specific telerehabilitation exercises targeted at FOG. In addition, the NFOGQ has been shown to be unsuitable as an outcome in RCT [64]; thus, objective measures of FOG severity [65,66] should be considered in RCT.

Kwok et al. [25] showed that eight sessions (90 min each session) of tele-yoga improved the BBS scores in eight people with moderate PD, but the absence of a control group makes it impossible to draw meaningful conclusions from this study. 

James-Palmer et al. [26] showed that although 12 sessions (30 min each session) of tele-yoga is safe and feasible for 16 people with mild-to-moderate PD, the intervention has no effect on the five times sit-to-stand performance. On the other hand, Pinto et al. [28] observed a significant improvement in the five times sit-to-stand of 12 people with mild-to-moderate PD after 16 sessions of dance. Although these studies did not use a control group for comparison, dance intervention is feasible for PD and it may improve the lower-limb performance of people with PD.

Kwok et al. [25] showed that eight sessions (90 min each session) of tele-yoga decreased the MDS-UPDRS-III score in eight people with moderate PD. Bianchini et al. [10] showed that 5 sessions, supervised by PT, plus 10 sessions, self-conducted (30 min each session), of mobility and postural transition exercises decreased the MDS-UPDRS-III score in 23 people with mild PD severity. James-Palmer et al. [26] showed that although 12 sessions (30 min each session) of tele-yoga are safe and feasible for people with mild-to-moderate PD (*n* = 16), the intervention does not affect the MDS-UPDRS-III score. It is important to highlight that the MDS-UPDRS-III score, excluding rigidity and postural stability items, was assessed remotely in these studies [10,25,26]. However, the absence of a control group in these three studies makes it impossible to draw meaningful conclusions.

One case-report study showed positive changes in walking speed after 52 sessions (60 min each session) of treadmill, virtual-reality training by videoconferencing [19] in two people with moderate PD. Anghelescu et al. [8] showed that 10 sessions (50 min each session) of adapted physiotherapy exercises in standing and sitting positions improved walking speed (6 m walk) and mobility (TUG test) in 17 people with mild-to-moderate PD. Only one study verified the effects of telerehabilitation by videoconferencing using virtual reality (treadmill-virtual reality training) [19] on mobility assessed remotely. Previous systematic reviews and meta-analyses revealed a positive effect of virtual reality strategies on static and dynamic balance, walking speed, gait, and motor skills of people with PD [17,67]. However, studies do not show the superiority or inferiority of virtual reality over traditional motor rehabilitation, suggesting it can be used as an augmentation or prolongation of conventional rehabilitation [17,67]. In addition, most studies lack methodological quality, as observed in our scoping review, and as previously reported [17,67]. Future high-quality clinical trials should be performed to verify the effectiveness of telerehabilitation by videoconferencing with and without virtual reality training on the mobility of people with PD assessed remotely. Finally, all studies included in this scoping review show only the feasibility of different interventions in people with mild-to-moderate PD and the safety of assessing balance, walking, and mobility remotely. Thus, future powered studies with an experimental design (e.g., RCT) are needed to investigate the effectiveness of telerehabilitation by videoconferencing on balance and gait assessed remotely.

### 3.4. Effects of Telerehabilitation by Videoconferencing on Balance and Gait Assessed in Person

Of the 14 studies included in this scoping review (Table 1), 7 investigated the effects of telerehabilitation by videoconferencing on balance and gait assessed in person [16,57,58,59,60,61,62]. Balance was assessed through the Mini-BESTest in three studies [59,61,62], one study used the BESTest [58], three studies used the ABC scale [16,57,62], and one study used the BBS [16]. In the case study [59], 24 sessions of the home-based Baduanjin Qigong exercise program (stretching, breathing, seven low-intensity movements emphasizing static and dynamic postural control) increased the Mini-BESTest scores in two patients with moderate PD. In the non-RCT study [61], 16 sessions of home-based telerehabilitation through multimodal functional exercises improved the Mini-BESTest scores in 11 people with mild-to-moderate PD. In the RCT pilot study [62], 16 sessions of LSVT^®^ BIG were more effective than 16 sessions of Progressive Structured Mobility Training on Mini-BESTest scores in 17 people with mild-to-moderate PD. These results show that a telerehabilitation program consisting of large amplitude, functional movements (LSVT^®^ BIG) could be better than standardized mobility training in improving several aspects of balance (Mini-BESTest). Another RCT pilot study [58] showed that the BESTest scores increased similarly after 24 sessions of tango classes in both groups that performed tango classes in person (*n* = 10) and by videoconferencing (*n* = 10). These results suggest that a telerehabilitation approach of group tango classes for people with PD is feasible and may have similar outcomes to in-person instruction regarding static and dynamic balance.

Three PD telerehabilitation studies used the ABC scale as an outcome [16,57,62]. Although one RCT study showed that 36 sessions of motor telerehabilitation (e.g., mobility, strength, and balance training) do not affect the ABC scores of 8 people with mild PD [57], another RCT study observed improvements in ABC scores after 16 sessions of LSVT^®^ BIG compared to 16 sessions of Progressive Structured Mobility Training in 17 people with mild-to-moderate PD [62]. LSVT^®^ BIG involved large amplitude, functional movements which may have positively impacted the self-perception of balance control. ABC scores were also improved via telerehabilitation with exergames. One RCT study [16] showed that 21 sessions of home-based telerehabilitation by videoconferencing using the exergames (Nintendo Wii Fit system) (*n* = 38) with a caregiver to help the patient in conducting exercises are more effective than 21 sessions of in-clinic sensory integration balance training (*n* = 38) in improving BBS scores; on the other hand, in-person sensory integration balance training is more effective than the exergames in improving the Dynamic Gait Index. These results indicate that telerehabilitation by videoconferencing with exergames having a caregiver is a feasible alternative to in-person rehabilitation for reducing BBS scores, but facility-based training is superior in improving Dynamic Gait Index.

Walking was assessed through the 10 m walk test [16,59], 2 min walk test [59], 6 min walk test [61], 3 min walk test [62], and gait velocity normalized to leg length in both forward and backward directions during a 4.8 m walk using the GAITRite computerized walkway [58]. In the case study [59], 24 sessions of the home-based Baduanjin Qigong exercise program (stretching, breathing, seven low-intensity movements emphasizing static and dynamic postural control) improved the performance in the 10 m walk test and 2 min walk test in two patients with moderate PD. In the non-RCT study [61], 16 sessions of home-based telerehabilitation through multimodal functional exercises did not improve the performance in the 6 min walk test in 11 people with mild-to-moderate PD. One RCT pilot study [58] showed that the gait velocity in both the forward and backward directions did not improve after 24 sessions of tango classes in both groups that performed tango classes in person (*n* = 10) and by videoconferencing (*n* = 10). One RCT study [16] showed that the performance in the 10 m walk test improved similarly after 21 sessions of home-based telerehabilitation by videoconferencing using the exergames (Nintendo Wii Fit system) with a caregiver to help the patient in conducting exercises and in-clinic sensory integration balance training. Another RCT study observed similar improvements in the gait speed, double step length, and TUG of people with mild-to-moderate PD after 16 sessions of either LSVT^®^ BIG or Progressive Structured Mobility Training applied by videoconferencing and supervised by PT [62]. Only the two last RCTs showed improvement in walking [16,62] and mobility [62], following supervised telerehabilitation by videoconferencing [16,62] or facility-based training [16]. Although these results suggest that telerehabilitation by videoconferencing and facility-based training could have the same effect on walking performance, the positive effects of telerehabilitation by videoconferencing on walking scores are still unclear due to a small number of RCTs included in this scoping review. 

Motor symptoms were assessed through the MDS-UPDRS part III in one non-RCT study without a control group [60] and in two RCT studies [57,58]. Sixty sessions of physiotherapy by videoconferencing did not decrease the MDS-UPDRS-III of 22 people with mild PD severity [60]. Similarly, 36 sessions of motor telerehabilitation (e.g., mobility, strength, and balance training) did not decrease the MDS-UPDRS-III score of eight people with mild PD [57]. Only one RCT study showed that the MDS-UPDRS-III score improved similarly after 24 sessions of tango classes in both groups that performed tango classes in person (*n* = 10) and by videoconferencing (*n* = 10). These results indicate that tango classes by videoconferencing are a feasible alternative to in-person rehabilitation for reducing MDS-UPDRS-III score, although these results should be confirmed in future studies with a more robust methodology.

## 4. Discussion

### 4.1. Effects of Telerehabilitation by Videoconferencing on Balance and Gait Outcomes

This scoping review found that of the 14 studies of balance and gait telerehabilitation for PD by videoconferencing [8,10,16,19,25,26,27,28,57,58,59,60,61,62], 71.4% did not use an experimental design [8,10,19,25,26,27,28,59,60,61], leaving a gap in understanding the effects of telerehabilitation by videoconferencing on balance and gait. In addition, seven non-RCT trials [8,10,19,25,26,27,28] assessed and treated balance and gait by videoconferencing. The failure to use a control group and small sample sizes make it difficult to draw meaningful conclusions from these studies. For now, based on the studies available in the literature, our scoping review cannot recommend telerehabilitation by videoconferencing to improve balance and gait in people with PD. Future, better-powered randomized studies should investigate the effectiveness of balance and gait telerehabilitation by videoconferencing in improving balance and gait outcomes with an emphasis on assessing remotely as well.

Balance and gait telerehabilitation by videoconferencing for PD lacks strong evidence-based trials and it presents unique safety concerns [68] since we need to determine how patients can practice balance and gait exercises in the home during intervention with minimal risk for falls. Meta-analyses demonstrated that training at facilities led to more improvement in balance and gait [5], and a decrease in motor symptoms (UPDRS-III and MDS-UPDRS-III) [69] in people with PD over the long term compared to independent, community, and home-based training. Facility-based training, supervised by the physical therapist, enables participants to practice performance at their optimal capacity since participants can practice challenging and complex exercises with more intensity and less fear of falling [5]. This physical therapist-led training can implement the clinical Practice Guideline’s recommendation to improve balance and gait [70], such as motor learning principles and task specificity that target specific impairments in PD such as anticipatory postural adjustment, sensory integration, reactive stepping, and complex gait to improve balance and gait in people with PD [70]. At this point, it is unclear whether we can adequately implement these recommendations to improve balance and gait remotely in a safe way. 

An important consideration in designing and implementing appropriately dosed exercises to improve gait and balance is that they may unfortunately lead to falls during telerehabilitation by videoconferencing, particularly for people with moderate-to-severe PD. Only one study in this scoping review included patients in H&Y stage 4 or those with cognitive impairments [27]. In this study, although most exercises were delivered in a seated position, one participant reported sustained injuries (low-back pain) and another fell during exercises, but no medical intervention was required. Other studies have reported the challenge of including people with mild-to-moderate PD in telerehabilitation programs to improve balance or prevent falls [16,61]. These studies have used balance exercises with the progressive challenge of postural control and balance in the presence of a caregiver, which have been demonstrated as a feasible alternative to in-clinic balance training for improving balance and postural control in people with PD [16,61]. There were no falls or other adverse health problems during the sessions since caregivers were always present [16,61]. Thus, future telerehabilitation studies designed to reduce postural instability and balance problems should consider the presence of a caregiver to monitor people with PD during challenging training sessions, warranting its safety.

### 4.2. Effects of Telerehabilitation by Videoconferencing on Balance and Gait Outcomes: How Can We Implement Objective Measures by Videoconferencing? 

Interestingly, only four RCT studies investigated the effects of telerehabilitation by videoconferencing on balance and gait, and all four studies assessed the outcomes in person [16,57,58,62]. These findings demonstrate that videoconferencing approaches for balance and gait assessments in telerehabilitation programs are still scarce in RCT studies. Also, gold-standard clinical assessments of balance, the Mini-BESTest [59,61,62] and the BESTest [58], were assessed only in person in the telerehabilitation by videoconferencing programs. Balance is particularly challenging to assess by videoconferencing because it is difficult to (1) challenge imbalance remotely (e.g., postural perturbations, unstable surface), (2) capture multiple domains of balance objectively, (3) use sophisticated equipment in people’s homes, and (4) keep patients safe like in-person supervised assessments. Objective measures of balance are lacking in telerehabilitation approaches by videoconferencing. Objective measures are vital since there are discrepancies between patient-reported outcomes and objective assessments [71], which can lead to the overestimation or underestimation of treatment outcomes [71]. A commentary paper [72] suggests the use of remote balance scales (e.g., Activities of Balance Confidence) and lower-limb assessments (30 s chair test, five times sit-to-stand, and 2 min step test) during and beyond COVID-19. However, these assessments do not capture multiple domains of balance (e.g., anticipatory postural adjustment, postural reactions, postural sway, trunk range of motion, and turning) and may not have enough challenge to evoke imbalance during remote assessment. Previous studies have demonstrated the feasibility of remotely assessing balance-related measures such as five times sit-to-stand [73], 360-degree rapid-turn-test [73], and motor symptoms using MDS-UPRDS part III [10,74]; additionally, a modified MDS-UPDRS Motor Score that excludes rigidity and postural instability has been demonstrated to be generally feasible for telehealth visits in PD [74,75,76]. To the best of our knowledge, no study has assessed multiple domains of balance remotely nor used technology for the quantitative assessment of the effects of telerehabilitation. 

New strategies to overcome these obstacles in telerehabilitation by videoconferencing are now being pursued. A potentially more definitive RCT has been developed by Silva-Batista et al. [77] with the TelePD trial (ClinicalTrials.gov (accessed on 27 March 2024), NCT05680597) to implement both the assessment and treatment of balance and gait remotely using telerehabilitation by videoconferencing that is physical therapist-supervised. The TelePD trial is designed to enroll 80 participants with mild-to-moderate PD and randomize them at a 1:1 ratio into receiving home-based balance exercises in either (1) physical therapist-supervised telerehabilitation in real-time (experimental group, *n* = 40) or (2) unsupervised exercises at home (control group, *n* = 40). Both groups receive 12 sessions of intervention with the same exercises. Silva-Batista et al. [77] has adapted the original Agility Boot Camp (ABC) program previously published [51,66,78,79] to be performed remotely with appropriate safety modifications. The ABC program was designed to target several underlying constraints on balance. The study will also explore and validate remote, versus in-person, outcome measures, using both wearable sensor-based measures of balance and gait remotely as well as established clinical measures and questionnaires. 

### 4.3. How Can We Implement Wearable Technology into Telerehabilitation Programs by Videoconferencing?

Over the last several years, studies have shown the feasibility of wearable technology for balance and gait treatment in neurological diseases. According to a previous meta-analysis, most studies chose the use of biomechanical sensors, such as pressure and inertial sensors, preferably placed under the patient’s feet, aiming to provide biofeedback during rehabilitation and to measure the ground reaction force generated by the body and to give feedback on weight-bearing or center of pressure during the gait cycle [80].

Effective rehabilitation is dependent on motor-learning principles and, as such, is experience-dependent and responds to intermittent feedback delivery that allows time for the integration of sensory information into movement [81]. So, wearable devices should be capable of modulating biofeedback according to these principles to provide an effective learning environment during rehabilitation. A previous review showed that the use of wearable equipment that allows for continuous data collection (e.g., rapid and low-cost prototyping devices) is exponentially growing [80]. This advance is accompanied by the development of sensor data processing techniques and data availability techniques [82], which are important steps for effective monitoring and assistance during the telerehabilitation of chronic neurological diseases such as PD. A meta-analysis [35] showed that feedback-based interventions, using wearable sensors, have shown promising results for gait and balance rehabilitation in different populations (e.g., PD, stroke, and frail older adults). 

A recent study from our group showed the feasibility of wearable technology used by researchers and physical therapists to provide feedback, in real time, based on objective balance and gait metrics during the in-person rehabilitation program for PD [37]. Mobility Rehab uses visual biofeedback that allows the researchers and physical therapists to select from among a variety of upper, and lower, body gait metrics during training. Our findings showed that one session of treadmill gait training with the Mobility Rehab system, using wearable Opal sensors attached to the feet, wrists, and sternum of people with PD has immediate effects on upper and lower body gait metrics (e.g., foot-strike angle, arm ROM, and lateral trunk ROM) [37]. Our group has recently demonstrated the feasibility and effectiveness of eight sessions of physical therapy combined with the Mobility Rehab system in improving gait speed and arm swing range-of-motion in an outpatient clinic for older adults with mobility disturbances [38]. Although there is a growing body of research dedicated to facilitating the adoption of wearable technology in rehabilitation clinical practice, implementing wearable devices in a home-based telerehabilitation context is still a new and challenging task. 

During telerehabilitation by videoconferencing, physical therapists observe patients’ mobility patterns and provide verbal and/or somatosensory feedback to improve their mobility. However, these methods are not optimal because the observation of balance and gait is subjective and depends on the expertise of the physical therapist [83]. In addition, patients use a webcam/computer, a mobile phone, or a tablet when engaging in the videoconferencing assessments and treatments from their home and some patients may not have adequate space in their homes, which may limit the view of their balance and gait. Thus, the implementation of wearable technology during telerehabilitation allows therapists to have an objective characterization of balance and gait impairments that are difficult to observe subjectively, such as reduced trunk motion, decreased foot clearance, postural sway, and excessive double support time. A recent systematic review reinforces that wearable technology is needed in telemedicine infrastructure, which includes telerehabilitation programs [84]. Thus, future research should focus on developing and implementing wearable technology that provides real-time biofeedback both for the treatment and accurate assessment of balance and gait during home-based telerehabilitation. 

## 5. Conclusions

Although telerehabilitation by videoconferencing is becoming increasingly common for people with PD, the non-randomized designs, small numbers of subjects, heterogeneity of the exercise protocols, and failure to use a control group make it impossible to draw meaningful conclusions from the current studies that both assessed and treated people with PD by videoconferencing. Thus, robust evidence of the effects of telerehabilitation by videoconferencing treatment with assessment of balance and gait was not found and requires future powered, prospective, and robust clinical trials. However, the effects of telerehabilitation by videoconferencing on balance and gait assessed in person are positive, although caution needs to be taken, as only four RCTs in people with PD were found. Future studies should implement objective measures of balance and gait by wearable technology during videoconferencing. 

## Figures and Tables

**Figure 1 geriatrics-09-00066-f001:**
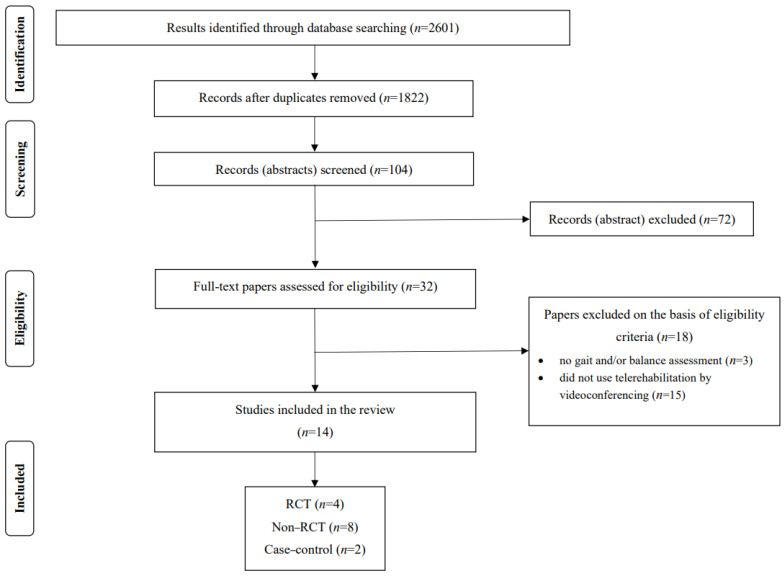
PRISMA flowchart of the included studies. RCT = randomized controlled trial.

**Table 1 geriatrics-09-00066-t001:** Summary characteristics of the included studies using telerehabilitation with videoconferencing.

Study and Country	Study	Participants	Experimental Group	Delivery Method	Control Group	Gait and Balance Outcomes	Gait and Balance Outcomes Assessed Remotely	Adverse Events	Author’s Conclusion
Anghelescu, 2022 [8] Romania	Non-RCT	*n* = 17 PD patients Age 65.9 ± 4.8 years, disease duration 7.3 ± 3.5 years, H&Y 1.5–3, men (*n* = 12), and women (*n* = 5)	Home-based motor telerehabilitation (toning, stretching, endurance, and balance exercises in sitting and orthostatic positions) 50 min/session, twice a week for 5 weeks	Real-time videoconference through laptop, smartphone, or tablet using Google Meet 2.1, Skype 3.1, or WhatsApp	None	TUG 6-Meter Walk	TUG 6-Meter Walk	Not reported	All patients improved their mobility, as there was a significant decrease in TUG duration. Telerehabilitation also significantly improved the average walking speed.
Bianchini et al., 2022 [10] Italy	Non-RCT	*n* = 23 PD patients Age 64.1 ± 8.9 years, disease duration 6.5 ± 3.8 years, H&Y 1–2.5, men (*n* = 13), and women (*n* = 10)	Home-based telerehabilitation program (general mobility, static, and dynamic balance, coordination, dexterity, postural transitions, and facial mobility). At least 30 min/session, 3 times/week for 5 weeks (a remote session with a physiotherapist once weekly and at least two self-conducted sessions per week)	Real-time videoconferencing + video tutorials showing exercises through the computer using the platform “Salute Digitale”	None	MDS-UPDRS III FIM	MDS-UPDRS III FIM	No adverse events occurred	The intervention is safe, feasible, and effective in reducing motor symptoms in mild-to-moderate PD patients.
Carvalho et al., 2021 [59] Canada	Case study	*n* = 2 PD patients Ages 75 and 74 years old, disease duration 12 and 17 years, H&Y 3, men (*n* = 1), and women (*n* = 1)	Home-based Baduanjin Qigong exercise program (stretching, breathing, seven low-intensity movements emphasizing static, and dynamic postural control) 3 times/week, for 8 weeks; 2 supervised sessions and 1 unsupervised session per week	Real-time videoconference through laptop using TeraPlus^®^ software 1.0 (a clinical information system with videoconferencing components)	None	Mini-BESTest 10 m walk test (self-selected and fast pace) 2 min walk test	None	No adverse events were reported during the interventions. However, one patient had 2 falls over the training period that were not related to the intervention.	The intervention seems potentially effective to improve important markers of walking performance: self-selected and fast-paced gait speed, and static and dynamic balance in PD.
Cornejo Thumm et al., 2021 [19] Israel	Case study	*n* = 2 PD patients Age 46 and 67 years old, disease duration 17 and 15 years, H&Y 3, one man, and one woman	Home-based, supervised, virtual reality telerehabilitation on a treadmill with a safety harness 15 to 60 min/session, once a week for 12 months	Real-time videoconferencing through computer, using Google Chrome 2.1 remote desktop tool and Skype 3.1 and treadmill-virtual reality	None	Gait Speed Walking endurance ABC scale MDS-UPDRS	Gait Speed Walking endurance ABC scale	No adverse events occurred.	The intervention is feasible. There was an improvement in gait speed, training endurance, and confidence in mobility, and disease symptoms presented a minor progression over the 12-month intervention period.
Gandolfi et al., 2017 [16] Italy	RCT	Experimental group: *n* = 38 PD patients, age 67.4 ± 7.2 years old, disease duration 6.2 ± 3.8 years, H&Y 2.5, men (*n* = 23), and women (*n* = 15) Control group: *n* = 38 PD patients, age 69.8 ± 9.4 years old, disease duration 7.5 ± 3.9 years, H&Y 2.5–3, men (*n* = 28), and women (*n* = 10)	Home-based, supervised, virtual reality telerehabilitation 50 min/session, 3 times/week for 7 weeks	Exergaming through Wii console using a balance board plus a computer connected with a high-resolution web camera for real-time remote visual communication via Skype 3.1.	Facility-based sensory integration balance training (SIBT) 50 min/session, 3 times/week for 7 weeks	ABC scale BBS DGI 10 m Walking test	None	No adverse events were reported during the study period.	Results show that static and dynamic postural control improved in PD patients from EG, while improvements in mobility and dynamic balance were greater in those from the CG. Similar effects on perceived confidence in performing ambulatory activities, gait speed, fall frequency, and quality of life were achieved in both groups.
Garg et al., 2021 [60] India	Non-RCT	*n* = 22 PD patients, age 66 (44–71) years, disease duration 4.9 ± 3.7 years, H&Y 1–2.5, men (*n* = 13), and women (*n* = 9)	Home-based, semi-supervised, telerehabilitation program 30 min/session, 5 times/week for 12 weeks. Supervised session once a week for the first 4 weeks, once every 2 weeks for the last 8 weeks	Real-time videoconferencing through smartphone (for supervised sessions) + handouts of different therapeutic exercises	None	MDS-UPDRS III	None	Not reported	The intervention was feasible but showed no significant effects on motor or non-motor symptoms of PD patients.
James-Palmer & Daneault, 2022 [26] USA	Non-RCT	*n* = 16 PD patients Age 63.1 ± 10.3 years old, H&Y 1–3, disease duration 4.8 ± 5.2 years, men (*n* = 6), and women (*n* = 10)	Home-based yoga exercises via telerehabilitation (breathing, yoga positions, relaxation) 30 min/session, twice a week for 6 weeks	Real-time videoconference through a laptop, smartphone, or tablet using Zoom 3.1	None	MDS-UPDRS-III (excluding rigidity and postural stability evaluations) FTSTS	MDS-UPDRS-III (excluding rigidity and postural stability evaluation) FTSTS	Mild adverse events not related to the intervention included baseline pain, new pain related to outside activity, not feeling well, unrelated losses of balance, and medication side effects.	The intervention is safe and feasible for people with mild-to-moderate PD. There were no significant differences in motor symptoms.
Kaya Aytutuldu et al., 2024 [62] Turkey	RCT	Experimental group: *n* = 17 PD patients, age 58.4 ± 8.2 years old, disease duration 4.8 ± 3.8 years, H&Y 2–3, men (*n* = 12), and women (*n* = 4) Control group: *n* = 17 PD patients, age 61.2 ± 6.7 years old, disease duration 6.6 ± 4.2 years, H&Y 2–2.5, men (*n* = 12), and women (*n* = 4)	Telerehabilitation using LSVT^®^ BIG protocol. 60 min/session, 4 times/week for 4 weeks	Real-time videoconference using Zoom 3.1.	Progressive structured mobility training 60 min/session, 4 times/week for 4 weeks	ABC-SF Mini-BESTest TUG Spatiotemporal gait parameters through Kinovea^®^	None	Not reported.	Both groups improved dynamic balance, postural stability, gait parameters, activity balance confidence, and activity status. However, dynamic balance, balance confidence, and activity status improvements favored the LSVT^®^ BIG group.
Kwok et al., 2022 [25] Hong Kong	Non-RCT	*n* = 8 PD patients Age 63.1 ± 5.4 years old, H&Y 3, disease duration not informed, men (*n* = 4), and women (*n* = 4)	Home and group-based mindfulness yoga training via telerehabilitation 90 min/session, twice a week, for 4 weeks	Real-time videoconference through laptop using Zoom	None	BBS MDS-UPDRS-III ABC scale-ON and OFF FOGQ	BBS MDS-UPDRS-III ABC-ON and OFF FOGQ	No adverse events occurred.	Results showed that the intervention was feasible, safe, and well accepted among people with PD. Participants showed a significant improvement in BBS, MDS-UPDRS-III, from baseline to 1-week follow-up.
Lavoie et al., 2021 [61] Canada	Non-RCT	*n* = 11 PD patients Age 69.2 ± 3.6 years old, H&Y 2–3, disease duration 8.4 ± 3.9 years, men (*n* = 6), and women (*n* = 5)	Home-based telerehabilitation through multimodal functional balance and flexibility exercises 60 min/session, twice a week for 8 weeks followed by unsupervised exercise 60 min/session, 3 times/week for 12 weeks	Real-time videoconferencing through TeraPlus^®^ 2.0 connected to controllable wide-angle pan-tilt-zoom cameras.	None	Mini-BESTest TUG 6MWT, MDS-UPDRS,	None	No adverse events occurred.	The intervention improved the dynamic balance of participants. The change in distance walked in the 6MWT was less than MDC for the test.
Pastana Ramos et al., 2023 [57] Brazil	RCT	Experimental group: *n* = 8 PD patients, age 60.7 (49–72) years old, disease duration 5 (3–9) years, H&Y 1–2, men (*n* = 4), and women (*n* = 4) Control group: *n* = 11 PD patients, age 58.6 (53–64) years old, disease duration 4 (2–11) years, H&Y 1–2, men (*n* = 6), and women (*n* = 5)	Home-based telerehabilitation program including mobility, strength, and balance exercises. 60 min/session, 3 times/week for 12 weeks	Real-time videoconferencing through smartphone, laptop, or tablet using free teleconference platforms (e.g., Google Meet^®^ 2.1)	Received booklet with demonstration and description of exercises from telerehabilitation program and were instructed to perform exercises 3 times/week at home.	ABC scale FTSTS MDS-UPDRS-III TUG	None	Only 3 minor adverse events related to intervention were reported (2 presented pain and 1 tiredness).	No significant differences were observed between telerehabilitation and control groups.
Pinto et al., 2023 [28] Brazil	Non-RCT	Experimental group: *n* = 12 PD patients, age 69 (65.2–72.8) years old, disease duration 8.6 (4.5–12.7) years, H&Y 1–4, men (*n* = 2), women (*n* = 10), freezers (*n* = 6), and non-freezers (*n* = 6) Control group: *n* = 14 older adults, age 69 (64.6–73.3) years old, men (*n* = 1), and women (*n* = 13)	Home-based dance sessions (75% seated) developed from the Dance for PD^®^ materials, including aspects of ballet, modern dance, jazz, tap, samba, forró, salsa, and tango. 60 min/session, twice a week for 8 weeks	Real-time videoconferencing using Zoom.	Same as experimental group.	ABC scale FTSTS	ABC scale FTSTS	No adverse events occurred	Time to perform FTSTS only decreased in the PD group. Balance confidence (ABC scale) diminished in the PD group.
Seidler et al., 2016 [58] USA	RCT	Experimental group: *n* = 10 PD patients, age 68.1 ± 7.9 years old, disease duration 4 (2–10) years, H&Y 2–3, men (*n* = 4), women (*n* = 6), freezers (*n* = 4), and non-freezers (*n* = 6) Control group: *n* = 10 PD patients, age 68.9 ± 9.4 years old, disease duration 2.3 ± (1.4–7.8) years, H&Y 2- 2.5, men (*n* = 5), women (*n* = 5), freezers (*n* = 4), and non-freezers (*n* = 6)	Facility-based group tango dance classes with a teleconferenced instructor 60 min/session, twice a week for 12 weeks	Real-time videoconferencing through a laptop connected to webcams and a projector using Acrobat Connect.	Facility-based, group tango dance classes with a face-to-face instructor 60 min/session, twice a week for 12 weeks	BESTest MDS-UPDRS-III Forwards and backwards gait velocity (GAITRite)	None	No adverse events occurred.	Telerehabilitation tango dance was a feasible intervention and produced similar improvement in balance and motor signs outcomes compared to face-to-face dance classes in PD patients.
Tardelli et al., 2022 [27] Brazil	Non-RCT	Experimental group: *n* = 57 PD patients, age 66.9 ± 9.8 years old, disease duration 7.6 ± 5.2 years, H&Y 2.6 (1–4), men (*n* = 30), women (*n* = 27), freezers (*n* = 21), and non-freezers (*n* = 36) Control group: *n* = 29 PD patients, age 65.1 ± 9.9 years old, disease duration 8 ± 5.7 years, H&Y 2.8 (2–4), men (*n* = 15), women (*n* = 14), freezers (*n* = 9), and non-freezers (*n* = 20)	Home-based, real-time telerehabilitation (2 sessions of sitting and standing dance activities, one session of sitting and standing physical therapy) 60 min/session, 2–3 times/week for 10 months	Real-time videoconference through laptop, smartphone, or tablet using free software (e.g., Google Meet 2.1 and Skype 3.1)	Non-exercising	Walking and posture (items 28 and 29 of UPDRS-III) NFOG-Q	Walking and posture (items 28 and 29 of UPDRS-III) NFOG-Q	One participant reported sustained low-back pain) for three weeks while performing stationary walking with the dual task. Another participant fell while performing chest-press with an elastic band with a high resistance level. No medical intervention was required.	The intervention is more effective than non-exercising control in preserving walking in people with mild-to-moderate PD who were frequent exercisers before the pandemic, although it does not positively affect the subjective posture and FOG.

Abbreviations: RCT = randomized controlled trial; PD = Parkinson’s disease; H&Y = Hoehn & Yahr stage; ABC scale = Activities-Specific Balance Confidence scale; BBS = Berg Balance Scale; BESTest = Balance evaluation systems test; DGI = Dynamic Gait Index; FIM = Functional Independence Measure; FTSTS = Five times sit-to-stand test; MDS-UPDRS-III = Movement Disorders Society Unified Parkinson Disease Rating Scale section for motor impairment; NFOG-Q = New Freezing of Gait Questionnaire; 6MWT = Six Minutes Walking Test; TUG = Timed-up-and-Go test; UPDRS-III = Unified Parkinson’s Disease Rating Scale-motor subscale.

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
