# Peer review of "Telerehabilitation by Videoconferencing for Balance and Gait in People with Parkinson’s Disease: A Scoping Review"

_geriatrics, 2024, doi:10.3390/geriatrics9030066_

Round 1

Reviewer 1 Report

Comments and Suggestions for Authors

Modern technologies are used not only in clinical medicine but also in long-term health care. Therefore, this work is important and may be the reason for the increased interest of researchers dealing with this topic.

Author Response

Modern technologies are used not only in clinical medicine but also in long-term health care. Therefore, this work is important and may be the reason for the increased interest of researchers dealing with this topic.

Response: Thank you so much.

Reviewer 2 Report

Comments and Suggestions for Authors

The idea and the outcome of the study is clear. I found some minor problems.  Lines 115-120: 2601 - 1822 makes 779 (not 104). One of the counts is wrong. In addition, in the text "Of these, 72 were excluded based on the title and abstract. Of the remaining 72 articles, 32 were fully assessed according to the inclusion criteria..." How can it be? After exclusion of 72 from 104 only 32 are reamining. Should it be "The remaining 32 articles were fully..."?

Author Response

The idea and the outcome of the study is clear. I found some minor problems.  Lines 115-120: 2601 - 1822 makes 779 (not 104). One of the counts is wrong. In addition, in the text "Of these, 72 were excluded based on the title and abstract. Of the remaining 72 articles, 32 were fully assessed according to the inclusion criteria..." How can it be? After exclusion of 72 from 104 only 32 are reamining. Should it be "The remaining 32 articles were fully..."?

Response: Thank you so much. Thanks for catching that, we have improved the sentence as follows:

  1. Results

3.1. Study Selection

To investigate the current evidence on the effects of telerehabilitation on balance and gait, delivered by videoconferencing for PD, the search identified 2601 articles retrieved from computerized databases up to September 2023 (PubMed, ISI’s Web of Knowledge, Cochrane’s Library, and Embase). After removing duplicate items (779), 1822 remained in which titles and abstracts were read and 104 were selected. Of these, 72 were excluded and 32 were fully assessed for eligibility; however, only 14 studies met the study criteria, which were included in the final analysis. The search process is depicted in Figure 1.

Reviewer 3 Report

Comments and Suggestions for Authors

First of all, I would like to thank the journal editor for the opportunity to review the paper.

As the authors indicate in the introduction, Parkinson's disease is affecting an increasing number of individuals in the population, especially in developed countries where life expectancy is higher. Analyzing the scientific evidence on the use of teleassistance is undoubtedly interesting and topical. In this regard, the objective of the paper is justified.

Lines 53-56. Undoubtedly, telerehabilitation can provide services to many patients, especially those living in rural areas, but I suggest the authors make some reference to the potential cost savings that the use of this technology may entail for both public and private healthcare systems.

Lines 61-62. In my opinion, I consider that a more cautious approach should be taken in this statement. Currently, this limitation exists, but it is foreseeable that it will be overcome in a short time with technological advances and devices controlled by artificial intelligence. I believe this aspect should be referenced more systematically in the introduction.

I have concerns regarding the heterogeneity of the types of activity and physical exercise, as they are too disparate, as well as the number of sessions and sample sizes. In the paper, each of the studies reviewed is analyzed. In my opinion, rather than a discussion, it seems like a description of results. However, I understand the difficulty given the disparity in research characteristics. In my opinion, I suggest that the authors focus the discussion on the limitations of this narrative review.

The authors propose the following objective: "The study aims to review, with a narrative approach, the current evidence on the effects of telerehabilitation via videoconferencing on balance and gait for PD, as well as to explore whether studies have used wearable technology in telerehabilitation programs for both assessment and treatment via videoconferencing." I believe the objective should be rephrased. In my opinion, the first part is perfect, but the second part is not very clear. and, on the other hand, to adjust it according to the recommendations I make in the following paragraph

In my opinion, the conclusions should focus on referring to each of the sections or issues raised in the discussion. Specifically, 1.- Effects of telerehabilitation by videoconferencing on balance and gait assessed by videoconferencing 2.- Effects of telerehabilitation by videoconferencing on balance and gait assessed in-person, and in the conclusion as well, reference should be made to the "future perspectives" section.

Author Response

First of all, I would like to thank the journal editor for the opportunity to review the paper. As the authors indicate in the introduction, Parkinson's disease is affecting an increasing number of individuals in the population, especially in developed countries where life expectancy is higher. Analyzing the scientific evidence on the use of teleassistance is undoubtedly interesting and topical. In this regard, the objective of the paper is justified.

Response: Thank you so much.

Lines 53-56. Undoubtedly, telerehabilitation can provide services to many patients, especially those living in rural areas, but I suggest the authors make some reference to the potential cost savings that the use of this technology may entail for both public and private healthcare systems.

Response: Thank you, we have included a sentence as follows:

Two systematic reviews have investigated the costs associated with telerehabilitation [22, 23]. Kairy et al. [22] demonstrated that lower costs for healthcare arose from telerehabilitation. Del Pino et al. [23] found that telerehabilitation can be less costly and burdensome than in-person rehabilitation performed in the clinic for people with neurological and cardiovascular diseases. Although telerehabilitation delivered remotely shows lower costs compared with face-to-face rehabilitation at home [24], it is still unknown if telerehabilitation is as effective and safe as facility-based rehabilitation, especially for balance and gait impairments.

Lines 61-62. In my opinion, I consider that a more cautious approach should be taken in this statement. Currently, this limitation exists, but it is foreseeable that it will be overcome in a short time with technological advances and devices controlled by artificial intelligence. I believe this aspect should be referenced more systematically in the introduction.

Response: We have changed this sentence as follows:

Successful telerehabilitation programs by videoconferencing may require the ability to also assess balance and gait outcomes remotely, although we have found no evidence in the literature that the same balance and gait outcomes, assessed remotely and in person, provide the same information. Balance and gait are difficult to assess by videoconferencing because it is not clear we can evoke imbalance remotely (e.g., postural perturbations, unstable surfaces) and capture multiple domains of balance and gait objectively, while keeping, patients safe like in-person, supervised assessments.

I have concerns regarding the heterogeneity of the types of activity and physical exercise, as they are too disparate, as well as the number of sessions and sample sizes. In the paper, each of the studies reviewed is analyzed. In my opinion, rather than a discussion, it seems like a description of results. However, I understand the difficulty given the disparity in research characteristics. In my opinion, I suggest that the authors focus the discussion on the limitations of this narrative review.

Response: Thank you so much for your suggestions. We have changed the order of titles and subtitles (see below) and focused on the limitations in the discussion.

  1. Results

3.1. Study Selection

3.2. Study Characteristics

3.3. Effects of telerehabilitation by videoconferencing on balance and gait assessed by videoconferencing

3.4. Effects of telerehabilitation by videoconferencing on balance and gait assessed in-person

  1. Discussion

4.1. Effects of telerehabilitation by videoconferencing on balance and gait outcomes

4.2. Effects of telerehabilitation by videoconferencing on balance and gait outcomes: How we can implement objective measures by videoconferencing?

4.3. How can we implement wearable technology into telerehabilitation programs by videoconferencing?

  1. Conclusions

The authors propose the following objective: "The study aims to review, with a narrative approach, the current evidence on the effects of telerehabilitation via videoconferencing on balance and gait for PD, as well as to explore whether studies have used wearable technology in telerehabilitation programs for both assessment and treatment via videoconferencing." I believe the objective should be rephrased. In my opinion, the first part is perfect, but the second part is not very clear. and, on the other hand, to adjust it according to the recommendations I make in the following paragraph

Response: We have made it clearer as described in the abstract:

Therefore, our study aimed to review, with a narrative approach, current evidence on the effects of telerehabilitation by videoconferencing on balance and gait for PD. We also explored whether studies have used wearable technology in telerehabilitation programs in both assessment and treatment by videoconferencing.

In my opinion, the conclusions should focus on referring to each of the sections or issues raised in the discussion. Specifically, 1.- Effects of telerehabilitation by videoconferencing on balance and gait assessed by videoconferencing 2.- Effects of telerehabilitation by videoconferencing on balance and gait assessed in-person, and in the conclusion as well, reference should be made to the "future perspectives" section.

Response: Thanks, we have changed it as follow:

Although telerehabilitation by videoconferencing is becoming increasingly common for people with PD, the non-randomized designs, small numbers of subjects, heterogeneity of the exercise protocols, and failure to use a control group make it impossible to draw meaningful conclusions from the current studies that both assessed and treated people with PD by videoconferencing. Thus, robust evidence of the effects of telerehabilitation by videoconferencing treatment with assessment of balance and gait was not found, and requires future powered, prospective, and robust clinical trials. However, the effects of telerehabilitation by videoconferencing on balance and gait assessed in-person are positive, although caution needs to be taken, as only 4 RCTs in people with PD were found. Future studies should implement objective measures of balance and gait by wearable technology during videoconferencing.